# Leukocyte differentiation in bronchoalveolar lavage fluids using higher harmonic generation microscopy and deep learning

**Laura M. G. van Huizen**[1,☯]*, **Max Blokker**[1,☯], **Yael Rip**[1], **Mitko Veta**[2], **Kirsten A. Mooij Kalverda**[3], **Peter I. Bonta**[3], **Jan Willem Duitman**[3,4,5], **Marie Louise Groot**[1]

**1** LaserLab Amsterdam, Department of Physics, Faculty of Science, Vrije Universiteit Amsterdam, Amsterdam, The Netherlands, **2** Medical Image Analysis Group (IMAG/e), Department of Biomedical Engineering, University of Technology, Eindhoven, The Netherlands, **3** Department of Pulmonary Medicine, Amsterdam UMC Location University of Amsterdam, Amsterdam, The Netherlands, **4** Department of Experimental immunology, Amsterdam UMC Location University of Amsterdam, Amsterdam, The Netherlands, **5** Amsterdam Infection & Immunity, Inflammatory Diseases, Amsterdam, The Netherlands

☯ These authors contributed equally to this work.

* l.m.g.van.huizen@vu.nl

**Data Availability Statement:** All data, model, code files are available from the DataverseNL database (accession number DOI: 10.34894/7NHFCL).

## Abstract

### Background

In diseases such as interstitial lung diseases (ILDs), patient diagnosis relies on diagnostic analysis of bronchoalveolar lavage fluid (BALF) and biopsies. Immunological BALF analysis includes differentiation of leukocytes by standard cytological techniques that are labor-intensive and time-consuming. Studies have shown promising leukocyte identification performance on blood fractions, using third harmonic generation (THG) and multiphoton excited autofluorescence (MPEF) microscopy.

### Objective

To extend leukocyte differentiation to BALF samples using THG/MPEF microscopy, and to show the potential of a trained deep learning algorithm for automated leukocyte identification and quantification.

### Methods

Leukocytes from blood obtained from three healthy individuals and one asthma patient, and BALF samples from six ILD patients were isolated and imaged using label-free microscopy. The cytological characteristics of leukocytes, including neutrophils, eosinophils, lymphocytes, and macrophages, in terms of cellular and nuclear morphology, and THG and MPEF signal intensity, were determined. A deep learning model was trained on 2D images and used to estimate the leukocyte ratios at the image-level using the differential cell counts obtained using standard cytological techniques as reference.

### Results

Different leukocyte populations were identified in BALF samples using label-free microscopy, showing distinctive cytological characteristics. Based on the THG/MPEF images, the

**Funding:** This publication is part of the project InstantPathology (with project number 15825) of the research program Applied and Engineering Sciences which is (partly) financed by the Dutch Research Council (NWO), awarded to M.G. The funders had no role in study design, data collection and analysis, decision to publish, or preparation of the manuscript.

**Competing interests:** I have read the journal's policy and the authors of this manuscript have the following competing interests: M.G. declares to have financial and non-financial interest in Flash Pathology B.V. However, Flash Pathology B.V. was not involved in the design of the study or analysis of the data. This does not alter our adherence to PLOS ONE policies on sharing data and materials.

deep learning network has learned to identify individual cells and was able to provide a reasonable estimate of the leukocyte percentage, reaching >90% accuracy on BALF samples in the hold-out testing set.

## Conclusions

Label-free THG/MPEF microscopy in combination with deep learning is a promising technique for instant differentiation and quantification of leukocytes. Immediate feedback on leukocyte ratios has potential to speed-up the diagnostic process and to reduce costs, workload and inter-observer variations.

## Introduction

Leukocytes are part of the immune system and are key players in the defense against infectious agents and injuries, for example in case of inflammation and tissue repair [1]. Most abundant leukocytes in the human body are neutrophils, eosinophils, lymphocytes, and monocytes/macrophages, but also other subtypes such as plasma cells and basophils are recognized. In many diseases the leukocyte composition is indicative of the underlying disease. Interstitial lung diseases (ILD) comprise a broad variety of lung diseases that affect the lung interstitium. In absence of a single diagnostic gold standard test, diagnosis of ILD relies on a multidisciplinary team (MDT) taking all available data in consideration to establish a working diagnosis with the highest possible diagnostic confidence. In this regard, immunological bronchoalveolar lavage fluid (BALF) analysis for leukocyte composition and quantification is part of the diagnostic work-up in ILDs, next to more invasive additional tissue sampling such as lung (cryo) biopsy [2–4]. Currently, analysis of BALF cell populations relies on labor-intensive and time-consuming cytological techniques by well-trained personnel.

A promising imaging technique for label-free rapid on-site feedback is third harmonic generation (THG) and multiphoton excited autofluorescence (MPEF) microscopy, in short label-free microscopy. This technique provides images of fresh specimens revealing histology information without processing the tissue [5–9]. Previous studies using blood samples have shown that THG signals reveal discriminative information for the different leukocytes, providing cellular and nuclear morphological information [10–12]. The THG signal intensity has been shown to be higher for granulocytes (neutrophils and eosinophils) than agranulocytes (lymphocytes and monocytes) because granulocytes contain THG generating granules in their cytoplasm [10, 11]. Cell cytoplasm also generates MPEF signals which differ between the various leukocyte in both fluorescence pattern and intensity [13, 14]. For example, the cytoplasmic granules of eosinophils generate strong autofluorescent signals. Previous studies have focused on cell differentiation using label-free microscopy in blood samples, while to the best of our knowledge, THG and MPEF microscopy has not been applied to BALF.

Automatic identification and classification of the different leukocyte types in the label-free images could potentially provide rapid feedback on BALF. Manual counting of leukocytes in these images is time consuming, and prone to interobserver variations. When only image-level annotations are available, the task of leukocyte counting is a perfect candidate for deep learning to undertake. Deep learning allows development of cell counters based on the image data. Studies have shown that the label-free images are suitable for automatic image analysis using deep learning networks [15–17]. These studies have used deep learning networks to classify malignant tissue from healthy tissue and to classify activated lymphocytes, but not for cell

counting. Automated cell counting based on U-net segmentation was demonstrated on label-free stimulated Raman scattering images of human brain tumors [18]. Regression based counting generates cell counts based directly on the entire input image, therefore avoiding the need to annotate cells beforehand. Deep learning regression was performed on microscopy images of human osteosarcoma and human leukemia cell lines to count cells [19], where they used absolute cell counts for only one cell type to train the model.

We present a proof-of-principle study with the goal to characterize and quantify different immune cells (neutrophiles, eosinophiles, lymphocytes and macrophages) in label-free THG/MPEF images of BALF samples. First, we characterized the four most frequent leukocytes populations in label-free microscopy images of blood fractions. Subsequently, we extended the leukocyte identification to BALF samples and show, for the first time, that identification of different leukocyte populations using label-free microscopy can be performed successfully. Additionally, for automatic differential immune cell counting, we trained a deep learning model on both blood fractions and BALF 2D images with as reference the corresponding standard cytology differential cell counts. Here we present a proof-of-principle deep learning model that provides immune cell ratios of label-free microscopy BALF images with >90% accuracy.

## Materials and methods

### I. Sample acquisition, preparation, and processing

Two types of samples were used to address the aim of this study, i.e. blood samples and bronchoalveolar lavage fluid (BALF) samples (Fig 1A). Samples were collected at the Amsterdam UMC (location AMC) and imaged with the label-free microscope directly on-site or in case the mobile label-free microscope was at Amsterdam UMC location VUmc, samples were

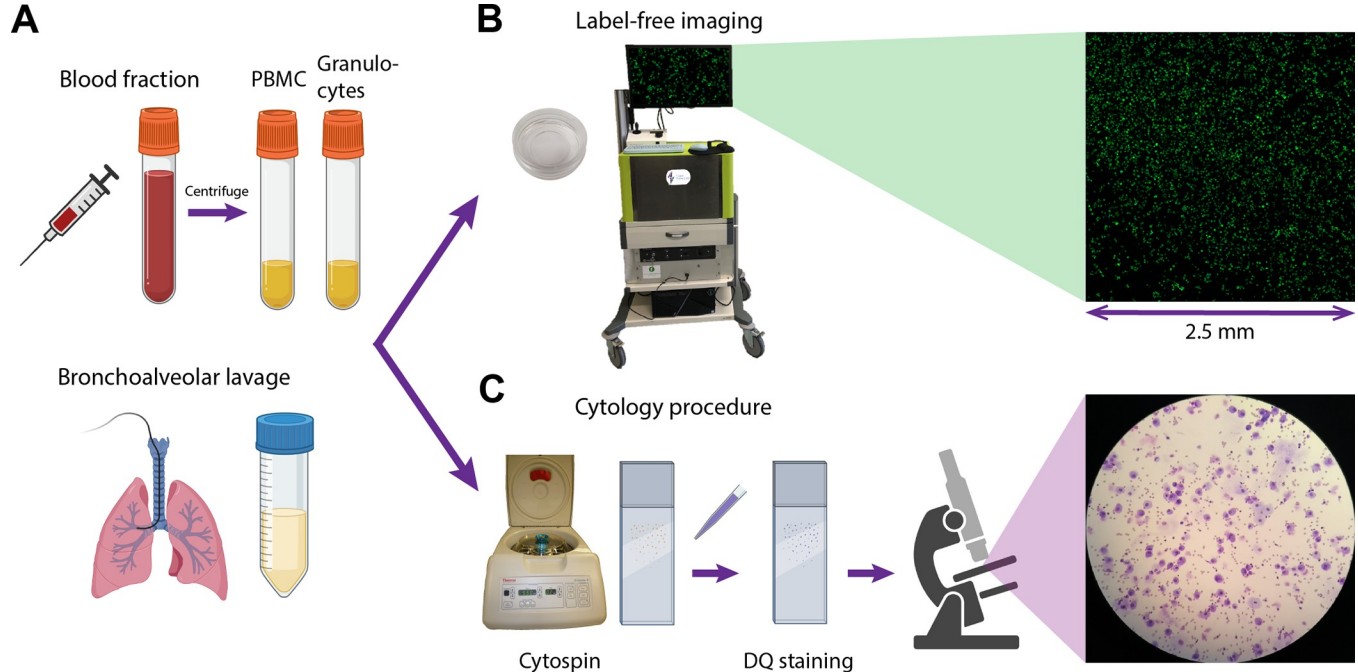

**Fig 1.** Schematic representation of sample acquisition (A), label-free imaging using a THG/MPEF microscope (B) and standard cytology processing using Diff-Quick (DQ) staining (C). Images partly created with BioRender.com.

**Table 1. Overview of the imaged samples, acquired number of mosaics and total imaged areas.**

| Sample type | ID | Patient condition | Number of samples | Total mosaics | Total imaged area |
|---|---|---|---|---|---|
| Granulocytes | GR 1–3 | Healthy | 3 | 4 | 15.63 mm$^2$ |
| | GR 4 | Asthma | 1 | 1 | 6.25 mm$^2$ |
| PBMC | PBMC 1–3 | Healthy | 3 | 3 | 18.75 mm$^2$ |
| Bronchoalveolar lavage | BAL 1–6 | ILD | 6 | 11 | 68.75 mm$^2$ |

Samples include blood fractions (granulocytes and PBMC) of four volunteers and BALF samples of six ILD patients. Of the person with asthma only the granulocyte fraction was included. The average number of immune cells per mm$^2$ for granulocyte fractions: 4808 cells, for PBMC fractions: 6464 cells, and for BALF samples: 549 cells.

transported on ice to the Imaging Centre at Amsterdam UMC, location VUmc and imaged within three hours. All patients provided written informed consent as part of studies approved by a Medical Research Board under protocol number NTR NL7634 or FP02C-18-001. The research was conducted in accordance with the Netherlands Code of Conduct for Research Integrity and the Declaration of Helsinki.

Six BALF samples were obtained from ILD patients who underwent bronchoalveolar lavage (BAL) during an endoscopic procedure. Samples were collected by installing 8 fractions of 20 mL saline solution, of which of the last fraction the last few milliliters were collected separately for label-free imaging. Samples were immediately cooled on ice after the bronchoscopy procedure, transported to the lab and processed according to the standard clinical procedures. Venous blood (three healthy volunteers and one asthma patient) was collected in heparin tubes and peripheral blood mononuclear cells (PBMCs) and granulocytes were isolated using standard density gradient techniques. From healthy volunteers both PBMC and granulocyte fractions were included, and from the asthma patient only the granulocyte fraction. See Table 1 for an overview of included samples. Subsequently, samples (BALF and blood cell fractions) were processed according to standard cytology protocols of the Long Immunology (LONI) lab of Amsterdam UMC location AMC, resulting in Diff-Quick (DQ, RAL Diagnostics, ref 720555–000) cytospin slides (Fig 1C). Differential cell counts (macrophages/monocytes, lymphocytes, neutrophils, basophils, and eosinophils) were analyzed by counting at least 500 cells using a standard light microscope at 40× magnification, by two independent experienced observers. Results were calculated as percentages of the total cell population (Table 2) and considered as the reference standard values of these samples.

A few milliliters of each unprocessed sample/fraction were transferred to a sample holder dish (μ-Dish 35 mm, high, Ibidi GmbH). After a waiting time of at least 15 minutes to let the cells rest and sink to the bottom of the dish, the cells were imaged with the label-free microscope (Fig 1B).

## II. Data acquisition and image processing

To image the samples a portable higher harmonic generation (HHG) microscope was used (Flash Pathology B.V.). Details of the microscope have been described in our previous study [8]. In short, this microscope focuses a femtosecond-pulsed laser beam with a center wavelength of 1070 nm into the sample, which generates nonlinear multiphoton signals, relevant in this study: third harmonic generation (THG), two-photon excited autofluorescence (2PEF) and three-photon excited autofluorescence (3PEF). These three signals are detected in epi-direction and separated on wavelength using appropriate dichroic mirrors and filters at 349–361 nm (THG), 562–665 nm (2PEF), and 380–420 nm (3PEF). Because of practical reasons not all signals could be detected at the same time, and the measurement was performed twice subsequently, once with

**Table 2. Reference cytology differential cell percentages and standard deviation per sample.**

| Sample type | | ID | Neutrophils (%) | Eosinophils (%) | Lymphocytes (%) | Monocytes / macrophages (%) |
|---|---|---|---|---|---|---|
| Blood fraction | Granulocytes | GR1 | 94.5 (± 0.3) | 5.0 (± 0.5) | 0.5 (± 0.3) | 0.0 (± 0.0) |
| | | GR2 | 94.0 (± 1.3) | 5.0 (± 0.5) | 1.0 (± 0.8) | 0.0 (± 0.0) |
| | | GR3 | 88.3 (± 1.9) | 10.6 (± 1.1) | 1.1 (± 0.8) | 0.0 (± 0.0) |
| | | GR4 | 69.2 (± 0.6) | 14.1 (± 0.4) | 14.4 (± 0.6) | 2.3 (± 0.3) |
| | PBMC | PBMC1[a] | 0.4 | 0.0 | 98.1 | 1.5 |
| | | PBMC2 | 1.9 (± 0.8) | 0.0 (± 0.0) | 94.6 (± 0.6) | 3.5 (± 1.3) |
| | | PBMC3[a] | 2.9 | 0.6 | 95.3 | 1.2 |
| Bronchoalveolar lavage | | BAL1 | 5.1 (± 0.6) | 0.3 (± 0.1) | 35.3 (± 2.0) | 59.3 (± 1.5) |
| | | BAL2 | 17.5 (± 2.3) | 4.2 (± 0.1) | 4.7 (± 0.8) | 73.5 (± 2.9) |
| | | BAL3 | 92.7 (± 2.8) | 1.1 (± 0.2) | 1.8 (± 0.5) | 4.4 (± 2.3) |
| | | BAL4 | 32.0 (± 0.9) | 4.2 (± 0.5) | 6.4 (± 0.6) | 57.4 (± 1.0) |
| | | BAL5 | 0.3 (± 0.1) | 0.1 (± 0.1) | 4.4 (± 0.8) | 95.2 (± 0.8) |
| | | BAL6 | 3.0 (± 0.1) | 10.7 (± 0.9) | 13.8 (± 2.0) | 72.5 (± 1.0) |

Training set samples shaded in white, validation set samples in light gray, testing set samples shaded in dark gray.

[a] For this fraction no standard deviation is provided since the fraction was counted by one observer.

detecting THG and 2PEF signals simultaneously and once with THG and 3PEF signals. The microscope has a lateral resolution of 0.4 μm and axial resolution of 1.5 μm.

By scanning the laser beam with galvo mirrors over the sample an image was acquired with a field-of-view of 250 μm, and 2 or 4 pixels per micrometer, depending on the scan program. Subsequent images were stitched into a mosaic image (2D) by moving a translational stage in x- and y-direction, and into a stacked depth scan (3D) by moving the stage in z-direction.

For each granulocyte fraction, PBMC fraction and BALF sample one or two stitched mosaic scans and various stacked depth scans were acquired using the label-free microscope (Table 1). Each image was gamma corrected (factor 0.7) and saved to a 24-bit RGB BMP file, using built-in software (Flash Pathology B.V.)

### III. Leukocyte characterization

The label-free images of the blood fractions were analyzed qualitatively to determine the leukocyte characteristics, except for macrophages which are absent in blood. Both the cell morphology and THG and MPEF (2PEF and 3PEF) signal intensities were compared. These characteristics included the cellular diameter, determined by measuring 100 cells per immune cell type using the software ImageJ (Fiji version 1.53). The macrophage characteristics were evaluated using the label-free images of BALF samples. Together with what is known from literature, a complete set of leukocyte characteristics were formed to identify the different leukocytes in the label-free images of the BALF samples.

### IV. Deep learning cell counting

For deep learning cell counting, the blood fraction and BALF cases were separated on patient level into a training, validation, and testing set (see Table 2). Each split contained at least one granulocyte, one PBMC and one BALF case. From each case, only 2D mosaics that consist of THG and 2PEF images were included. Given that each mosaic covered a large area of the respective sample, therefore including a large number of immune cells, each mosaic was treated as an individual case. Accordingly, every mosaic was labelled with the reference cytology percentages from their respective sample.

Each mosaic was processed using background subtraction and contrast limited adaptive histogram equalization (CLAHE). Background subtraction allowed for removal of glass interface interference in the THG channel, due to THG being an interface-sensitive technique, and cells localizing at the bottom of the imaging dish. CLAHE enhanced the contrast of the large area images, evenly revealing all the immune cell-information present in the image data. Pixel size was fixed across all images to 0.5 μm by down-scaling if necessary. Due to the small sample size, larger mosaics ($> 2500 \times 2500$ pixels) were divided into four mosaics and treated as individual cases. This was possible due to the large numbers of cells captured with our microscope, since we scanned regions up to multiple millimeters squared depending on the field-of-view of the scan program. This also alleviated the GPU memory constraints. In total, this resulted in 40 samples in the training set, 10 samples in the validation set, and 16 samples in the testing set.

The deep learning regression network was set up as a fully convolutional network (FCN) to allow for dynamic image input size while keeping the pixel size fixed. The underlying network architecture consisted of ResNet50 [20] pre-trained on ImageNet [21, 22]. In addition, EfficientNetV2B0 [23] and MobileNetV2 [24] were investigated. The last fully-connected layer was replaced with global average pooling to obtain a 2D vector, a 4-unit fully-connected layer and subsequent softmax activation. The network was optimized by minimization of the mean absolute error, using Adam [25] and one cycle learning rate method [26] for a duration of 200 epochs with an initial learning rate of $1 \times 10^{-5}$. The baseline ResNet50 model weights were frozen for the first epoch and unfrozen for the remaining 199 epochs. Early stopping was applied after 15 epochs without decrease of the validation loss.

Due to over- and underrepresentation of certain immune cells across the data, the neural network might be inclined to favor one class of immune cells over the other during optimization. Since the leukocyte counts are fractions which always sum to 1, the setup was equivalent to multi-class classification with class imbalance. The imbalance was combated by application of class weights in the calculation of the loss function during training. In addition, potential overfitting due to a small training dataset was countered by the application of data augmentation during training. Upon each epoch, each input image was randomly rotated, flipped, and brightness, contrast, saturation, and hue adapted.

The top-performing model, based on the performance on the validation set, was evaluated on the hold-out testing set. Each case in the testing set consisted of one mosaic; each mosaic was fed into the model and testing regression percentages were saved and compared to reference cytology leukocyte ratios.

Implementation of the deep learning cell counting was done in TensorFlow 2.10 [27], and training took place on a GPU workstation (Lambda Quad, Lambda Labs, Inc.).

## V. Deep learning class activation maps

The regression outputs of the ResNet50 model do not allow for further interpretation of the model performance. Localizing model outputs to objects (i.e. immune cells) is a major benefit for model validation. In order to validate deep learning regression outputs, gradient-weighted class activation mapping (Grad-CAM) was implemented [28] using the tf-explain plugin [29]. Grad-CAM uses gradient flow at the last convolutional layer of the model, extracting the gradients that contribute to the prediction of the class of interest, overlaying this as a heat-map on the original input image. In practice, Grad-CAM gives a rough estimate of the regions-of-interest in the input image that contribute the most to the class prediction. For each testing set mosaic, a Grad-CAM heatmap was generated for each immune cell class, resulting in four heatmaps per test mosaic. The deep learning cell counting pipeline is visualized in Fig 2.

## Results

### I. Label-free microscopy images of leukocytes

Four granulocyte fractions, three PBMC fractions and six BALF samples were included and imaged using the label-free microscope. Each mosaic was generated within, on average, three minutes. Despite the difficulty of imaging thin transparent samples in epi-direction, the individual cells were clearly visible in the THG/MPEF images (Fig 3 shows an example of a BALF sample). THG signals (depicted in green) showed morphological information with subcellular details and 2PEF and 3PEF signals (depicted in blue and purple, respectively) revealed the autofluorescence of cell cytoplasms, generated by the compounds flavin adenine dinucleotide (FAD) and nicotinamide adenine dinucleotide (NAD(P)H), respectively.

### II. Leukocyte characterization

To discriminate between the different leukocyte types, the label-free images of the granulocyte fractions and PBMC fractions were evaluated qualitatively and compared with literature findings. Fig 4A–4C shows various examples of the different immune cells visualized by the different imaging modalities (THG, 2PEF and 3PEF).

The clearest difference between the granulocyte and PBMC fraction was the higher THG signal intensity of the granulocyte fractions in comparison to the PBMC fractions. This is due

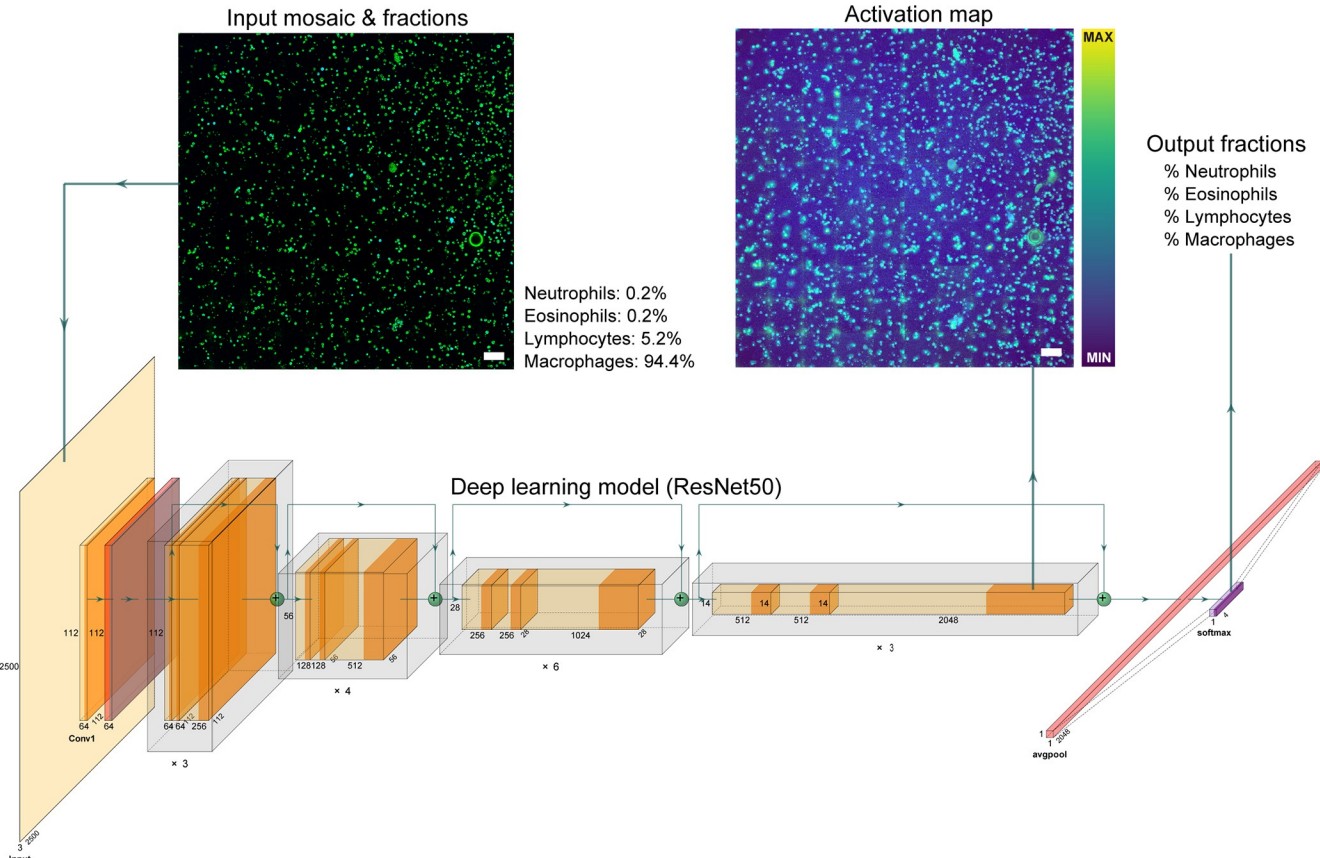

**Fig 2. Deep learning cell counting pipeline.** Pre-processed THG/MPEF mosaics are forward fed into the pre-trained ResNet50 model, together with reference cytology leukocyte ratios as labels. Class weights keep the model from overfitting on the ratio distribution. Mean absolute error is calculated between the fractions input and softmax output. After optimization, Grad-CAM activation maps can be extracted from the last convolutional layer for model validation. Scale bar 150 μm.

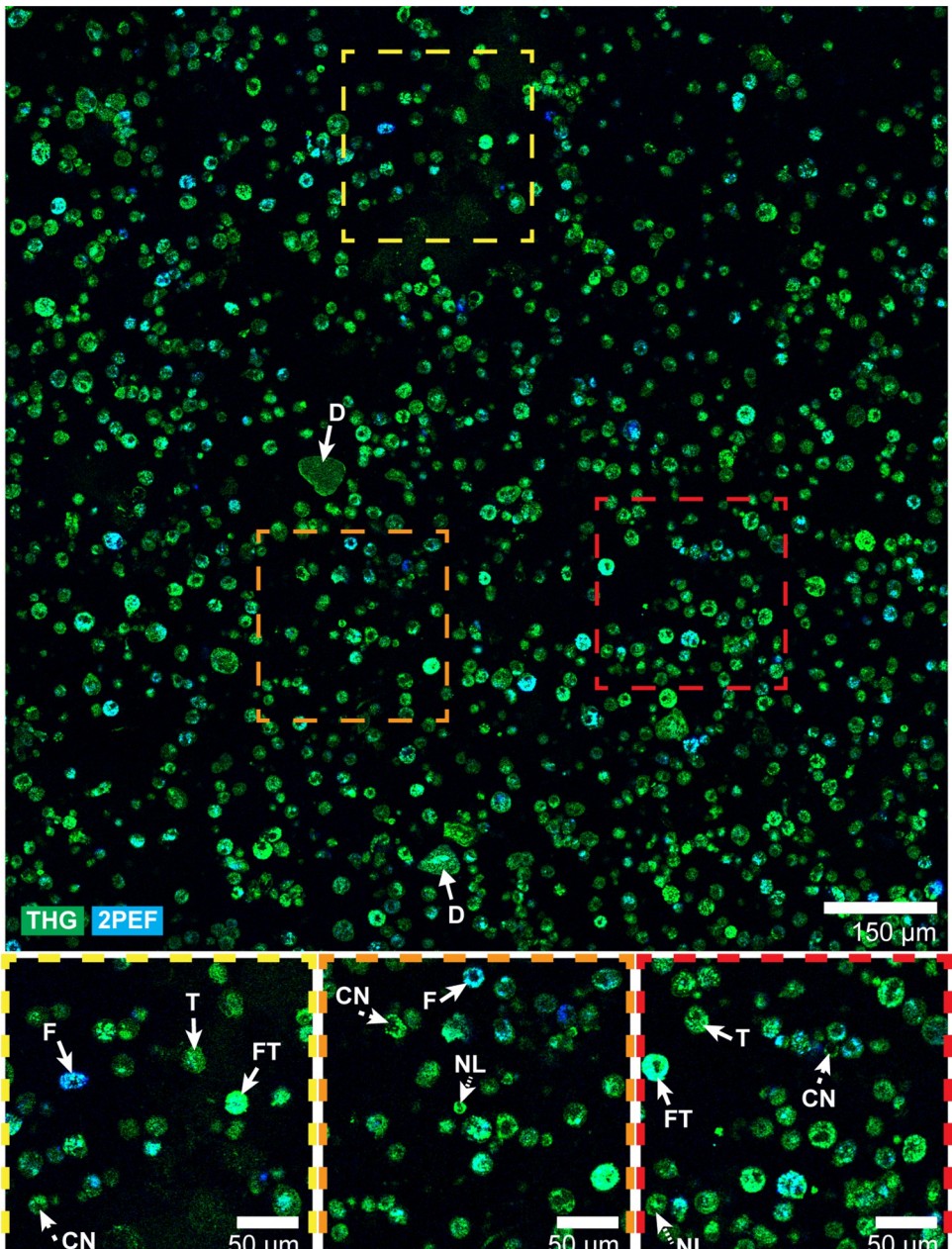

**Fig 3. Label-free imaging of a BALF sample showing cellular structures.** The BALF images contain cells of different sizes and shapes, and varying THG and 2PEF signal intensity. Magnified images show cell nuclei (CN), nuclear lobes (NL), and cell cytoplasm with increased 2PEF signal (F), THG signal (T) or both (FT). Non-cellular structures were also present, indicated as dirt (D). Acquisition time of this quarter of the acquired 2D mosaic was 38 seconds.

to strong THG signals generated by densely packed lipid granules inside the cytoplasm of granulocyte cells [10], while PBMC cells have no cytoplasmic granules. THG signals also revealed morphological differences between the different cell types, especially the cell size and the cell nuclear morphology are important discriminating factors. Lymphocytes were smaller in cell diameter, on average 7.1 µm (STD = 0.8 µm), compared to neutrophils and eosinophils with an average cell diameter of 8.9 µm (STD = 0.7 µm) and 8.7 µm (STD = 0.6 µm), respectively. Additionally, while the cell nuclei of lymphocytes and monocytes are mononuclear, the

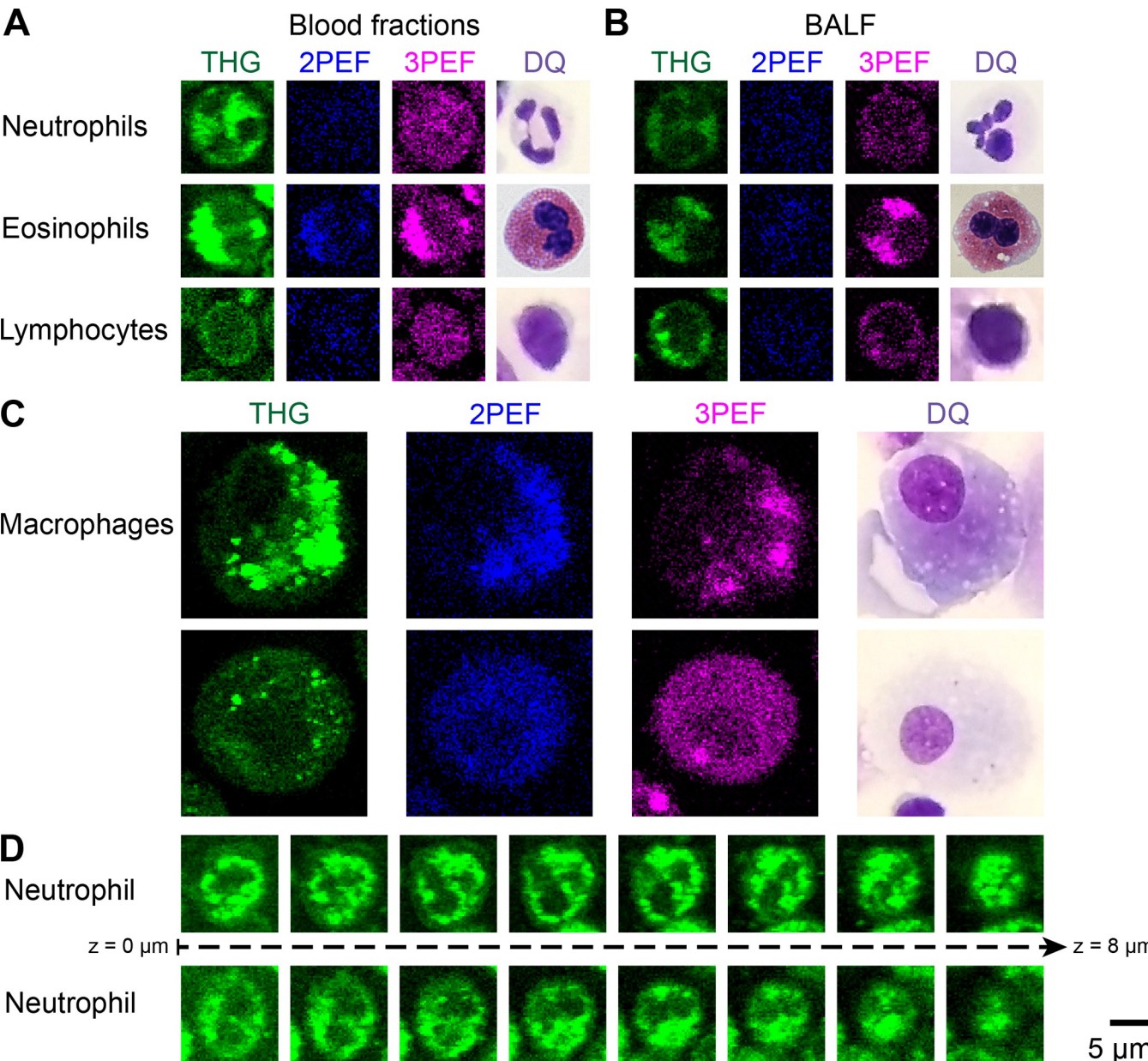

**Fig 4. Label-free images of different leukocytes showing the discriminating characteristics in cellular and nuclear morphology and signal intensity.** A-C: THG, 2PEF and 3PEF images of neutrophils, eosinophils, and lymphocytes from various blood fractions (A) and one bronchoalveolar lavage sample (B) show similar cell morphology as in standard cytology Diff-Quick (DQ) stained images. Neutrophils and eosinophils were imaged from the same granulocyte fraction and the images were saved with the same contrast settings. Two examples of macrophages are presented from two different BALF samples (C). D: 8 μm depth scans (with steps of 1 μm) of two neutrophils show that the visible number of nuclear lobes depends on the imaging plane. A-D: For all images the same scale was used, for direct comparison of cell sizes.

cell nuclei of neutrophils and eosinophils are multilobular, which is clearly visible in the THG images (Fig 4A). According to literature, which is inconclusive about the number of lobes, the neutrophils have 2 or 3 to 5 nuclear lobes [1, 30], while eosinophils usually have 2 or sometimes 3 nuclear lobes [1, 31]. However, in the 2D label-free images often only one or two lobes were visible, because of the position of the imaging plane, while in the 3D imaging stacks all lobes could be revealed (Fig 4D).

**Table 3. Leukocyte characteristics to distinguish the different cell types, based on THG and MPEF (2PEF and 3PEF) images.**

| Leukocyte type | Cell diameter[a] | Cell shape | Cell nucleus | Cell cytoplasm | THG signal | MPEF signal |
|---|---|---|---|---|---|---|
| Neutrophils | 8.9 µm (± 0.7) | Round | 2–5 lobes (not round) | Granules present | High | Low |
| Eosinophils | 8.7 µm (± 0.6) | Round | 2–3 lobes (not round) | Granules present | High | High |
| Lymphocytes | 7.1 µm (± 0.8)[b] | Round | Round/kidney-shaped mononuclear | No granules, little cytoplasm | Low | Low |
| Macrophages | 19.6 µm (± 3.0) | Varying | Round/kidney-shaped mononuclear[b] | No granules, can have pigment/ vesicles | Varying | Varying |

Discriminative features include average cellular size (measured of 100 cells per leukocyte type) and shape, nuclear morphology, the presence of granules and the THG and MPEF signal intensity.

[a] The average cell diameter of neutrophils, eosinophils and lymphocytes are measured in blood fractions, while of macrophages are measured in BALF samples

[b] Activated cells in BALF samples from ILD patients can differ in morphology, such as increased cell diameter up to 18 mm for lymphocytes, and multinuclear macrophages.

To discriminate within the granulocyte fraction between neutrophils and eosinophils, the MPEF images could be used, because the cell cytoplasm of eosinophils generated stronger 2PEF and 3PEF signals than the cell cytoplasm of neutrophils (Fig 4A). The 3PEF signals were stronger than the 2PEF signal, suggesting that NADH was mostly involved in generating the autofluorescence signals.

To determine the discriminative features of macrophages, and because macrophages are absent in blood, the BALF samples were evaluated. Fig 4C shows two examples. Similar to monocytes (the precursor cells of macrophages), macrophages are normally mononuclear and do not have granules in their cytoplasm. However, macrophages engulf and digest pathogens, and therefore their cytoplasm can contain inhomogeneities, such as pigments, that generate THG and MPEF signals. In contrast to neutrophils, eosinophils and lymphocytes that are round cells, macrophages were present in varying shapes. Also, the average diameter of macrophages of 19.6 µm (STD = 3.0 µm) was larger and varied more than of the other leukocytes.

Together, the cell size and shape, the nuclear shape, the presence of cytoplasmic granules, and the THG and MPEF signal intensities, form a complete set of characteristics to discriminate the different leukocytes (Table 3). Using these characteristics, all four cell types were identified in the BALF samples (Fig 4B, 4C).

## III. Deep learning cell counting

To estimate immune cell percentages automatically, a pre-trained ResNet50 was used to learn a regression task. Instead of absolute counts of the four different types of immune cells, the deep learning model had to learn class density dependencies in order to produce immune cell percentages. Several regression models were evaluated as part of the hyperparameter optimization process: ResNet50, EfficientNetV2B0, and MobileNetV2. Each network differs in architecture and in number of weights: 3.5 million (MobileNetV2), 7.2 million (EfficientNetV2B0) and 25.6 million (ResNet50), making it interesting to investigate regression performance between these models based on the validation set. In practice, the models performed similarly on the blood fraction samples in the validation set, while ResNet50 outperformed the other models on the BALF samples, which are of main interest. This might be because of the difficulty of the task at hand, requiring a larger model capacity to properly model the underlying correlations between image pixels and cytospin fractions. In addition to model architecture, the architecture weight initialization method, and learning rate were optimized. This demonstrated that pre-training on ImageNet and a learning rate of $1 \times 10^{-5}$ respectively resulted in the most optimal validation results. The training and validation metrics of the most optimal performing model are displayed in Fig 5, which also demonstrates the impact of class weighting on the loss function during training.

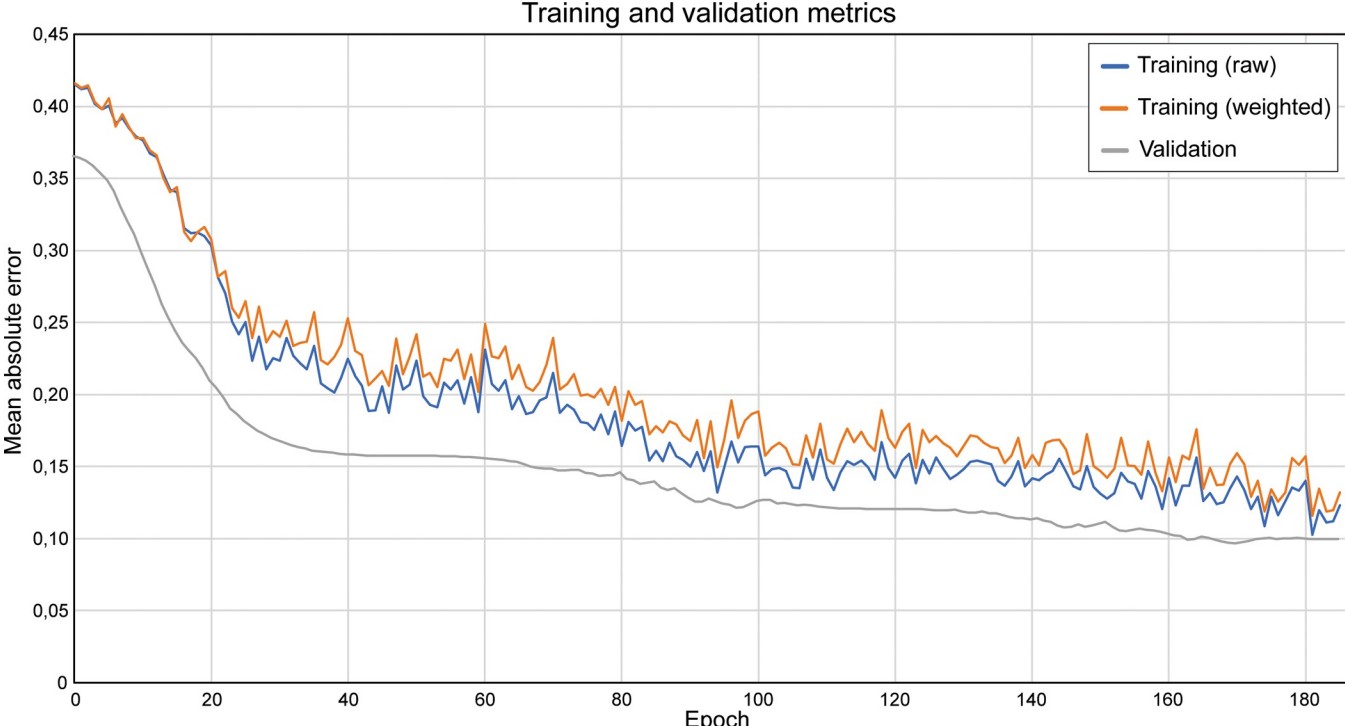

**Fig 5. Mean absolute error values of top-performing model during training and validation.** Graphs show mean absolute error (MAE) reported during training and validation. Solid blue line indicates pre-weighted training MAE values, solid orange line weighted training MAE values, and gray line validation MAE values. Weighting of the MAE takes place only during training by means of the class weights to avoid the model from overfitting on the leukocyte ratio distribution. As can be seen from the graphs, the class weights penalize raw MAE. The validation MAE is consistently lower than the training MAE, which could be explained by the lack of real-time data augmentation during validation.

In Fig 6 the top-performing deep learning model performance is compared to the reference cytology cell percentages for both the validation and testing set. Despite the model's difficulties to estimate leukocytes in blood fractions, it comes within a 2 to 10% error margin in BALF samples in both the validation and hold-out testing set. In general, the ResNet50 model tends to underestimate the neutrophil fraction, unless a very small number of neutrophils is present in the sample. The model makes up the underestimation by overestimating lymphocytes and monocytes/macrophages. For the BALF cases specifically, the model underestimates eosinophils.

Since a blood fraction/BALF case is made up of multiple mosaics carrying the same leukocyte fractions, a standard deviation of the model's predictions can be obtained for each case. These indicate that in the majority of the cases, the model performs regression with high certainty up to a standard deviation of 3%.

S1 Table lists the detailed fraction output of the top-performing ResNet50 model and reference cytology scores, for both the validation and testing sets.

## IV. Deep learning activation maps

Using Grad-CAM, a class activation map was generated for each immune cell type for test case BALF 4. Fig 7 displays zoomed-in activation maps for each immune cell class for this BALF case. An activation map indicates the contribution of each mosaic pixel to the class prediction. Between immune cell classes, the difference in pixel activation can be compared.

As shown in Fig 7, all cellular structures in the BALF mosaic are taken into account during class regression as evident from the yellow to red activations. For different classes, different

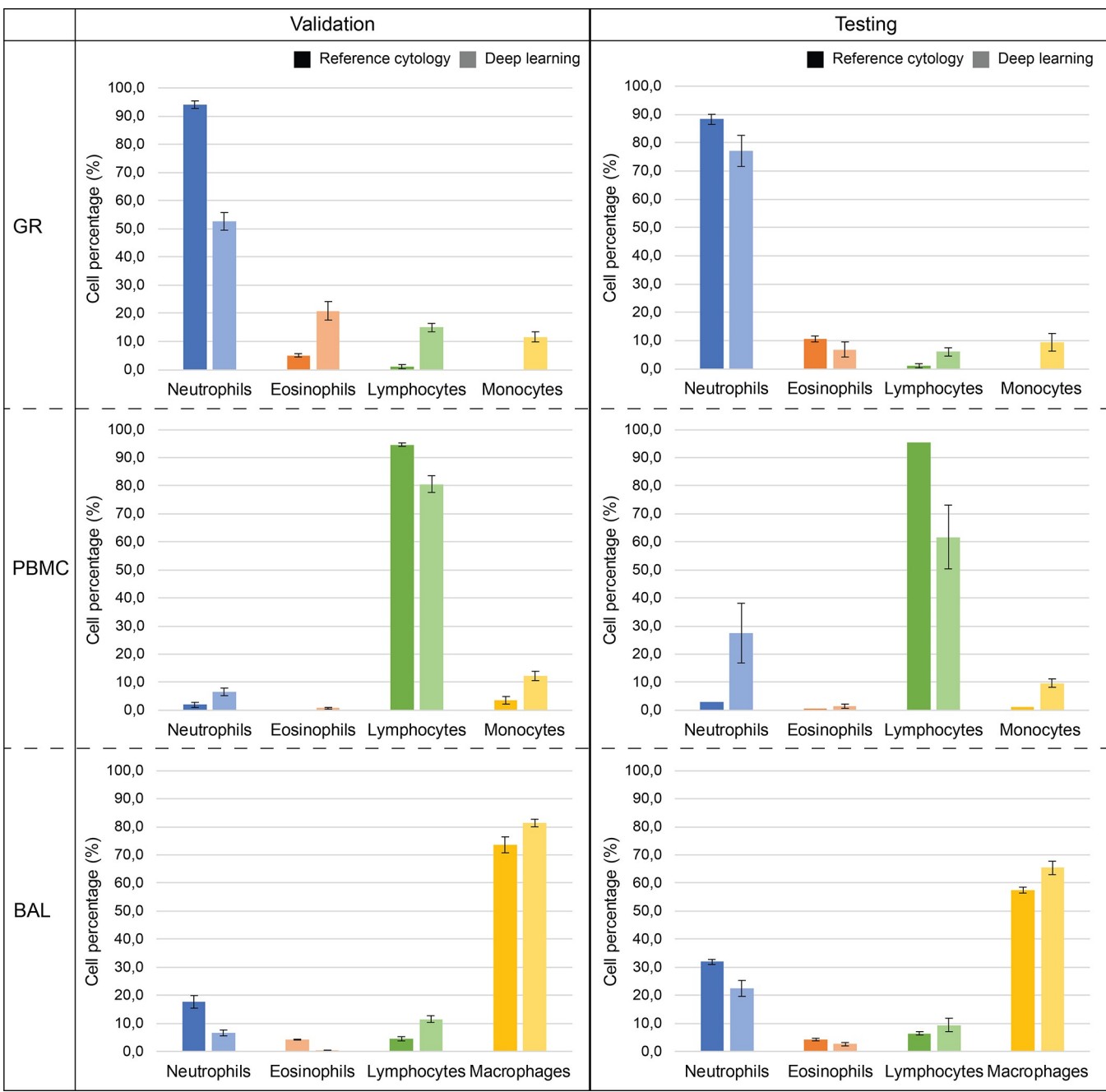

**Fig 6. Deep learning regression outputs on the validation and testing set compared to reference cytology counts.** See Table 2 for included cases in validation and test sets. Note that there is no standard deviation bar for the reference cytology of the PBMC in the testing set, because this sample was counted once.

maximum activations occur, but they seem to be mostly focused on single cells or clumps of cells. No direct correlation can be drawn between the number of low/high activated cells per class map and the leukocyte fractions. Similarly, a single cell might be activated in each of the four class activation maps, although different parts of the cell might be less or more activated depending on the class.

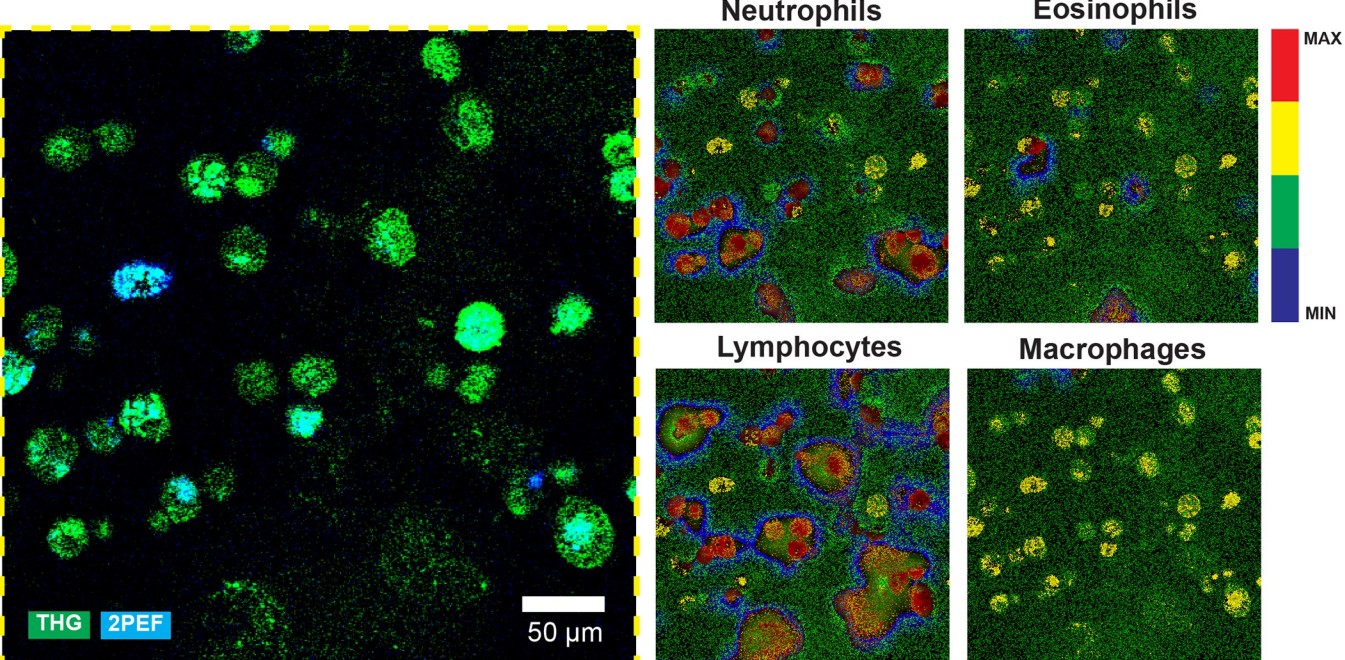

**Fig 7. Grad-CAM activation results for test case BALF 4.** See S1 Table for the regression performance of the ResNet50 model in this case. The activation maps are focused on the yellow square inlet from Fig 3 in the original mosaic image to illustrate individual cell activation. The maps are displayed using a ramping lookup table, which expresses the pixel value in four colors ranging from minimum to maximum activation: blue, green, yellow and red. Noticeable is the low activation of background pixels (green) and medium-to-high activation of cellular objects (yellow to red). Between class activation maps, the same part of a cell might be less or more activated. Scale bar 50 µm.

Another observation from Fig 7 is the clear distinction between background pixels and foreground pixels. However, the background pixels are still of interest to the model, given that they are colored green instead of blue. Cells with low signal, consisting of a cloud of pixels instead of solid objects, are still recognized and deemed of (high) importance to various class activations. Interestingly, high activations of cells are accompanied with a halo of low activations surrounding the cell, indicating a decision gradient with a clear distinction between background and foreground pixels.

## Discussion

This is the first study to image BALF samples using THG and MPEF (2PEF and 3PEF) microscopy. Even though the samples were transparent, and the generated signals were detected only in epi-direction by the microscope, the detected signal intensity was high enough to reveal the individual immune cells. In accordance with previous studies [10, 11, 13, 14] the different leukocytes could be easily identified in the blood fractions by cellular and nuclear morphology and distinctive THG and MPEF signal intensities. However, identification of the cells in BALF samples by human observers was more challenging. Additionally, we trained deep learning algorithms on image-level annotations only, based on reference cytology. The top-performing deep learning network trained was able to provide a leukocyte percentage estimation in substantial agreement with corresponding reference cytology percentages. In addition, class activation maps revealed that the model has learned to detect leukocytes without pixel-level annotations, relying only on leukocyte ratios.

In this proof-of-principle study we encountered some challenges. First, the 2D mosaics were acquired at a depth a few microns above the glass interface: close enough to the glass

surface to visualize the cells resting on the glass, including both large macrophages and small lymphocytes, but distant enough to avoid interference with the strong THG generated signals from the glass interface itself. As the intrinsic depth sectioning of the THG and MPEF was 1.5 micrometer, the visible cell size and cell nuclear appearance depends on the chosen imaging depth (as shown in Fig 4D). Depth scans, to reveal the full 3D information, could improve discrimination of the different leukocytes. Second, the reference cytology percentages were not counted from the exact same lung fluid as imaged with the microscope, but were from the same patients and lung area, which could have affected the reference leukocyte ratios. Instead of training on leukocyte ratios, an improvement would be to have a one-to-one comparison between cytology-stained slides and label-free images, with ideally annotated cells, which was unfortunately not feasible in this study. Third, it was more challenging to identify the leukocyte in BALF samples compared to blood fractions, because of the presence of more different cell types, and the overlapping characteristics of macrophages and the other leukocytes, especially in the 2D plane. Additionally, BALF samples were obtained from patients suspected of ILD, while blood fractions were mainly from healthy volunteers. Therefore, it is likely that in BALF samples activated immune cells were present that differ in appearance from healthy cells, both in cell morphology as in generated THG and MPEF signal intensities. For example, activated lymphocytes have increased cell sizes up to 18 μm [1] and generate stronger THG signals [12]. In addition, multinuclear macrophages can be present, called giant cells when the cell contains more than four nuclei. An atlas on how the different leukocytes can appear on the THG and MPEF images, from both inactive and active cells, from healthy and diseased patients, could help to discriminate the different leukocytes in BALF samples. Finally, because of practical reasons, only the THG and 2PEF images were included to train and test the deep learning network, and not the 3PEF images. The 3PEF images showed higher signal intensities from the cellular cytoplasm compared to the 2PEF images, and therefore it could be beneficial to include the 3PEF images in future studies.

The ultimate goal was to perform automatic leukocyte differentiation on BALF samples, utilizing blood fractions as supporting material. Looking at the obtained results in S1 Table and Fig 6, regression by deep learning based on the label-free images is feasible. With only 40 samples, annotated on the case-level by reference cytology fractions only, the algorithm learns to differentiate four different immune cell types and accurately detect cell density at least on the same order of magnitude as the current cytology workflow. While human observers are limited to training by a fixed set of cell morphology characteristics, the model has the freedom to extract any features that are necessary to achieve the same output fractions as the cytospin reference standard. The superior regression performance on BALF samples, compared to blood fractions, might be explained by the twice as many numbers of BALF samples available in our dataset. Another confounding factor could be the cell density difference between blood fraction and BALF, with the latter containing 10 times less cells for the same area (as shown in Table 1). Blood fraction data merely served as a support to increase the dataset, while the main focus is on BALF case differentiation and counting. The development and optimization of the deep learning model results in a fast analysis, faster than any counting process by a human observer. Training the ResNet50 until convergence took less than 10 minutes, leaving plenty of room for hyperparameter optimization. One reason for why training of the deep learning network took a short amount of time was the size of the training dataset which contained only 40 samples (12 blood fractions and 28 BALF samples) of 2500 × 2500 pixels sized mosaics.

When training on the raw, unprocessed (except for pixel size normalization) mosaics, the regression performance was similar to our results shown in this manuscript. However, upon inspection using Grad-CAM, most of the activations occurred on background pixels, glass-interface noise, and tile-by-tile stitching artefacts. Apparently, by focusing on background

space, the model was able to correlate background pixels to cell type densities. To mitigate, background subtraction was implemented using a rolling ball. Upon re-training and class activation visualization, the model was now focusing on cellular objects instead of background pixels. Still, the model is sensitive to intensity drops in mosaic tile corners, which was more enhanced in the validation blood fractions compared to the test samples. This might explain the regression performance drop in Fig 6.

The Grad-CAM atlases allow for a unique view into the model's decision process. They allow for thorough inspection of the gradients in the last convolutional layer of the ResNet50 model upon processing of a BALF test case mosaic. The downside of this approach is that it is unclear how much each pixel attributes to the actual model fraction output. Upon inspection of the activation maps, each class heatmap indicated that the model focused on cellular objects for the prediction of each class fraction. This underlines that each cell is of importance for the actual regression outcome of each class. This makes sense since for each leukocyte class, the first step is actual recognition of each single cell in the mosaic before moving on to counting only the cells of interest. In addition, cells can be activated for multiple leukocyte classes. This is an inherent flaw in the regression task design where we are unable to limit pixel attribution to a single class. Implementing a semantic segmentation strategy would solve this problem but would require pixel-level annotations. This is a general limitation in our setup where we do not have the manpower or training abilities to acquire annotations, which made the regression strategy a favorable one to start with.

Immediate feedback on immune cell content has potential for clinical guidance. Diagnosing ILD relies on careful assessment of all available data starting with non-invasive diagnostic tests, and lung biopsy is only considered in selected cases in which clinical-radiological and preferably, BALF results are unsupportive for a classifying diagnosis. Therefore, optimally BALF results will be awaited prior to the decision to proceed for biopsy, which implies that the patient will have to undergo a second bronchoscopy procedure in a separate session. Immediate interpretation of BALF during bronchoscopy using label-free microscopy could have potential to combine the BAL procedure and, upon indication, the biopsy procedure in a single session. Moreover, with a reliable automated analysis, diagnostic accuracy will be consistent for all procedures whereas with manual counting human error may arise when performed by less well-trained and experienced personnel. Additionally, identifying and discriminating leukocytes in the label-free images can aid identifying and discriminating of immune cells in label-free imaging of tissues, such as cryobiopsies obtained for ILD diagnosis. We are currently working on a study employing label-free imaging on fresh lung biopsies in ILD, where identifying the different leukocytes in the histology samples is important for ILD diagnosis. Moreover, considering the evolvement of THG/MPEF microscopy as an endo-microscope to evaluate tissue *in vivo*, identification of different cell types is crucial.

## Conclusions

In this proof-of-principle study we showed that using THG/MPEF microscopy neutrophils, eosinophils, lymphocytes, and macrophages can be identified in fresh, unprocessed BALF samples, showing distinctive characteristics. Additionally, we showed the capabilities of deep learning to perform regression on a limited and weakly-labelled dataset. In a next iteration, the performance could benefit from including more BALF samples of both various ILD diseased and healthy subjects. Furthermore, for future research we suggest a transition to 3D by acquisition of multi-layered mosaics across a depth of 10 to 20 μm to capture the full morphology of each cell. Considering the surmountable challenges discussed, we believe that THG/MPEF microscopy in combination with deep learning is a promising technique to differentiate

leukocytes in fresh, label-free imaged BALF samples, with the potential to speed-up the diagnostic process and reduce costs and workload across the board.

## Supporting information

**S1 Table. Evaluation of top-performing ResNet50 model on the validation set (top rows, light grey) and testing set (bottom rows, dark grey).** The average mean absolute error on the validation set was 0.136. The average mean absolute error on the testing set was 0.087. For each case in the two sets, the standard cytology percentages with standard deviation (except for the PBMC in the testing set, because this sample was counted once) are given and compared to the model's output. Given that a mosaic may be divided into multiple, single cases, an average regression output and standard deviation can be derived. For the blood fraction data, the macrophages class is replaced by monocytes, since monocytes are the precursor cells of macrophages present in blood.
(DOCX)

## Acknowledgments

We like to thank Tamara Dekker and Barbara Dierdorp for manually counting the leukocytes in the DQ cytospin slides.

## Author Contributions

**Conceptualization:** Laura M. G. van Huizen, Max Blokker, Kirsten A. Mooij Kalverda, Jan Willem Duitman, Marie Louise Groot.

**Data curation:** Laura M. G. van Huizen, Max Blokker.

**Formal analysis:** Laura M. G. van Huizen, Max Blokker.

**Funding acquisition:** Marie Louise Groot.

**Investigation:** Laura M. G. van Huizen, Max Blokker, Yael Rip.

**Methodology:** Laura M. G. van Huizen, Max Blokker, Jan Willem Duitman.

**Project administration:** Laura M. G. van Huizen.

**Resources:** Jan Willem Duitman.

**Software:** Laura M. G. van Huizen, Max Blokker.

**Supervision:** Mitko Veta, Marie Louise Groot.

**Validation:** Laura M. G. van Huizen, Max Blokker.

**Visualization:** Laura M. G. van Huizen, Max Blokker.

**Writing – original draft:** Laura M. G. van Huizen, Max Blokker.

**Writing – review & editing:** Laura M. G. van Huizen, Max Blokker, Mitko Veta, Kirsten A. Mooij Kalverda, Peter I. Bonta, Jan Willem Duitman, Marie Louise Groot.

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
