## [Decision Letter · Decision Letter 0]

14 Feb 2023

PONE-D-22-33842Differentiation of leukocytes in bronchoalveolar lavage fluid samples using higher harmonic generation microscopy and deep learningPLOS ONE

Dear Dr. van Huizen,

Thank you for submitting your manuscript to PLOS ONE. After careful consideration, we feel that it has merit but does not fully meet PLOS ONE’s publication criteria as it currently stands. Therefore, we invite you to submit a revised version of the manuscript that addresses the points raised during the review process. Please address the reviews as mentioned. There are some issues with image resolution. Provide training and loss graphs. Show a comparative analysis of how the model performs with and without adding weights to address class imbalance. 

We look forward to receiving your revised manuscript.

Kind regards,

Rahul Gomes, Ph.D.

Academic Editor

PLOS ONE

Journal Requirements:

2.Please provide additional details regarding participant consent. In the ethics statement in the Methods and online submission information, please ensure that you have specified what type you obtained (for instance, written or verbal, and if verbal, how it was documented and witnessed). If your study included minors, state whether you obtained consent from parents or guardians. If the need for consent was waived by the ethics committee, please include this information

"I have read the journal's policy and the authors of this manuscript have the following competing interests: M.G. declares to have financial and non-financial interest in Flash Pathology B.V. However, Flash Pathology B.V. was not involved in the design of the study or analysis of the data."

Reviewers' comments:

Reviewer's Responses to Questions

**Comments to the Author**

1. Is the manuscript technically sound, and do the data support the conclusions?

Reviewer #1: Yes

Reviewer #2: Yes

2. Has the statistical analysis been performed appropriately and rigorously? 

Reviewer #1: Yes

Reviewer #2: N/A

3. Have the authors made all data underlying the findings in their manuscript fully available?

Reviewer #1: Yes

Reviewer #2: Yes

4. Is the manuscript presented in an intelligible fashion and written in standard English?

Reviewer #1: Yes

Reviewer #2: No

5. Review Comments to the Author

Reviewer #1: -The motivation and background need to be improve and describe contribution and organization clearly in seprate paragraph in introduction section.

"A Numerical Study of Heat Transfer and Fluid Flow in a Channel with an Array of Pin Fins in Aligned and Staggered Configurations"

"Spectral control of high order harmonics through non-linear propagation effects"

-Author should uniform border in all tables.

-where is conclusion part? Author should add conclusion section and make sure not repeat phrases from the Introduction!.

-future work should more focused and clearly mention at the end of conclusion.

-please give a proofread check to the paper.

Reviewer #2: The title is very long, you can summare it

the inrorudcrion paer in the abstract section is long

you can highlight the main work in the intorudction section

The stucture of this paper can be added into the end of the intorudctions section

the mathmatical notation in this paper is missing

some related works are needed to be metioned in this paper

Artocarpus Classification Technique Using Deep Learning Based Convolutional Neural Network

Mango Varieties Classification-Based Optimization with Transfer Learning and Deep Learning ApproachesClassification Applications with Deep Learning and Machine Learning Technologies

6. PLOS authors have the option to publish the peer review history of their article (what does this mean?). If published, this will include your full peer review and any attached files.

Reviewer #1: No

Reviewer #2: No

---

## [Author Response · Author response to Decision Letter 0]

12 May 2023

Dr. Rahul Gomes

PLOS ONE

ATS Peer Review Office

1265 Battery Street, Suite 200

San Francisco, CA 94111

April 20, 2023

Subject: Submission of revised manuscript PONE-D-22-33842

Dear Dr. Rahul Gomes,

We thank the editors and reviewers for their time and constructive comments that have helped to improve our manuscript. We have carefully reviewed the comments and have revised the manuscript accordingly. Below, we provide a point-by-point response explaining how we have addressed each of the reviewers’ comments, corresponding changes to the text of the manuscript are marked in red.

Below you will also find the updated financial disclosure and competing interests statements:

Financial Disclosure statement

This publication is part of the project InstantPathology (with project number 15825) of the research program Applied and Engineering Sciences which is (partly) financed by the Dutch Research Council (NWO), awarded to M.G. The funders had no role in study design, data collection and analysis, decision to publish, or preparation of the manuscript.

Competing Interests statement

I have read the journal's policy and the authors of this manuscript have the following competing interests: M.G. declares to have financial and non-financial interest in Flash Pathology B.V. However, Flash Pathology B.V. was not involved in the design of the study or analysis of the data. This does not alter our adherence to PLOS ONE policies on sharing data and materials.

We hope the revised version is now suitable for publication and look forward to hearing from you in due course.

Yours sincerely,

On behalf of the co-authors,

PhD candidate Laura van Huizen,

PhD candidate Max Blokker 

and Prof. Dr. Marloes Groot

 

Response to editor:

C1: “There are some issues with image resolution”. 

R1: Thank you for pointing this out. We checked the resolution of our images, and our images are fulfilling the image requirements of the journal (format, dimensions, size, resolution). We agree that in the submitted PDF file the images have lower resolution. However, the single downloadable images have the correct quality. We hope you can appreciate the quality of these images, as the image resolution is important for this study.

C2: “Provide training and loss graphs.”

R2: We added a new figure, Figure 5, which displays the training and validation mean absolute error for the top-performing model during optimization.

C3: “Show a comparative analysis of how the model performs with and without adding weights to address class imbalance.”

R3: Thank you for raising this point. In the newly added Figure 5, this has been included. In short, the figure highlights the penalizing effect of the class weights on the raw training mean absolute error.

Response to Journal requirements:

C4: “Please ensure that your manuscript meets PLOS ONE's style requirements, including those for file naming.” 

R4: We made some small adjustments to the manuscript to meet all PLOS ONE’s style requirements, including placing the table legends at the bottom of the tables and adjusting the title page.

C5: “Please provide additional details regarding participant consent. In the ethics statement in the Methods and online submission information, please ensure that you have specified what type you obtained (for instance, written or verbal, and if verbal, how it was documented and witnessed). If your study included minors, state whether you obtained consent from parents or guardians. If the need for consent was waived by the ethics committee, please include this information”

R5: Thank you for pointing this out. We added the type of participant consent (written) to the ethics statement in Methods and online submission information, resulting in the full ethical statement (p. 6 line 124-127): “All patients provided written informed consent as part of studies approved by a Medical Research Board under protocol number NTR NL7634 or FP02C-18-001. The research was conducted in accordance with the Netherlands Code of Conduct for Research Integrity and the Declaration of Helsinki.”

C6: “Thank you for stating the following in the Competing Interests section:

"I have read the journal's policy and the authors of this manuscript have the following competing interests: M.G. declares to have financial and non-financial interest in Flash Pathology B.V. However, Flash Pathology B.V. was not involved in the design of the study or analysis of the data."

Please include your updated Competing Interests statement in your cover letter; we will change the online submission form on your behalf.”

R6: We adjusted the conflict-of-interest statement accordingly, resulting in the full Competing Interests statement: “I have read the journal's policy and the authors of this manuscript have the following competing interests: M.G. declares to have financial and non-financial interest in Flash Pathology B.V. However, Flash Pathology B.V. was not involved in the design of the study or analysis of the data. This does not alter our adherence to PLOS ONE policies on sharing data and materials.” 

Response to Reviewer #1:

Thank you for your review of our manuscript. We have answered each of your points below.

C7: “The motivation and background need to be improve and describe contribution and organization clearly in seprate paragraph in introduction section.”

R7: Thank you for your comments on the introduction. We improved the motivation and background of our study by simplifying and making them more concrete in the first paragraph (p. 4 line 63-75). Additionally, we added our contribution and the structure of the paper in the last paragraph of the introduction (p. 5-6 line 106-116). 

C8: “ "A Numerical Study of Heat Transfer and Fluid Flow in a Channel with an Array of Pin Fins in Aligned and Staggered Configurations"

"Spectral control of high order harmonics through non-linear propagation effects" “

R8: Thank you for the references. We assume the reviewer suggested these papers as references for in our manuscript. To our understanding the first paper describes the investigation of different configurations of array of pin fins for heat transfer and fluid flow experiments. We do not see how this paper connects to our manuscript, and therefore we did not include it as a reference. In the second mentioned paper the spectral shifts in generated harmonics in silicon and zinc oxide were investigated. Although this paper utilizes higher harmonic generation, similar as we do, we do not see any other overlap with our research, and therefore decided to not use it as a reference.

C9: “Author should uniform border in all tables.”

R9: Thank you for pointing this out. We adjusted all tables such that they have uniform borders and are fulfilling the requirements of the journal. 

C10: “where is conclusion part? Author should add conclusion section and make sure not repeat phrases from the Introduction!.”

R10: We agree with the reviewer that the conclusion part was not directly visible. We added the section “Conclusions” at the end of the manuscript, where we summarize the results (p. 23 line 530-540). 

C11: “future work should more focused and clearly mention at the end of conclusion.”

R11: Thank you for raising this point. In the new section “Conclusions” we added concrete suggestions for future work (p. 23 line 530-540). 

C12: “please give a proofread check to the paper.”

R12: Thank you for pointing this out. All authors have read the paper again carefully and we revised it to the best of our knowledge.

Response to Reviewer #2:

Thank you for reviewing and providing comments on our manuscript. We have answered each of your points below.

C13: “The title is very long, you can summare it”

R13: Thank you for pointing this out. We shorted the title to be: “Leukocyte differentiation in bronchoalveolar lavage fluids using higher harmonic generation microscopy and deep learning”. We believe that the components “leukocyte differentiation”, “BAL fluids”, “higher harmonic generation microscopy” and “deep learning” are important to be stated in the title, as together they form the focus of the manuscript. We are welcoming suggestions that also cover the content of the manuscript.

C14: “the inrorudcrion paer in the abstract section is long.”

R14: Thank you for raising this point. We shorted the introduction part of the abstract and provided more structure by adding headings (“background”, “objective”, “methods”, “results”, and “conclusions”). 

C15: “you can highlight the main work in the intorudction section”

R15: We agree with the reviewer that the main work can be highlighted more in the introduction section. We added, especially in the last paragraph of the introduction (p. 5-6 line 106-116), more concrete our contribution.

C16: “The stucture of this paper can be added into the end of the intorudctions section”

R16: We agree with the reviewer that the structure of the paper at the end of the introduction section is useful. Therefore, we outlined the structure of the manuscript in the last paragraph of the introduction (p. 5-6 line 106-116).

C17: “the mathmatical notation in this paper is missing”

R17: We assume that the reviewer is referring to “1e-05” in line 228 and 360 (old version). We replaced this by “1×10^(-5)” in line 228 and 358, respectively (revised version).

C18: “some related works are needed to be metioned in this paper

Artocarpus Classification Technique Using Deep Learning Based Convolutional Neural Network

Mango Varieties Classification-Based Optimization with Transfer Learning and Deep Learning Approaches

Classification Applications with Deep Learning and Machine Learning Technologies”

R18: Thank you for pointing out these references. We agree with the reviewer that the book containing these references is relevant given the similar models utilized in our work. Therefore, we have added a reference at p. 11 line 223.

R19: Reviewer #2 answered “No” to the following question: “Is the manuscript presented in an intelligible fashion and written in standard English?”. To the best of our knowledge, our manuscript is written in standard English in an intelligible fashion.

---

## [Decision Letter · Decision Letter 1]

12 Jun 2023

Leukocyte differentiation in bronchoalveolar lavage fluids using higher harmonic generation microscopy and deep learning

PONE-D-22-33842R1

Dear Dr. van Huizen,

We’re pleased to inform you that your manuscript has been judged scientifically suitable for publication and will be formally accepted for publication once it meets all outstanding technical requirements.

Kind regards,

Rahul Gomes, Ph.D.

Academic Editor

PLOS ONE

Additional Editor Comments (optional):

Reviewers' comments:

Reviewer's Responses to Questions

**Comments to the Author**

1. If the authors have adequately addressed your comments raised in a previous round of review and you feel that this manuscript is now acceptable for publication, you may indicate that here to bypass the “Comments to the Author” section, enter your conflict of interest statement in the “Confidential to Editor” section, and submit your "Accept" recommendation.

Reviewer #1: (No Response)

Reviewer #2: All comments have been addressed

2. Is the manuscript technically sound, and do the data support the conclusions?

Reviewer #1: (No Response)

Reviewer #2: Yes

3. Has the statistical analysis been performed appropriately and rigorously? 

Reviewer #1: (No Response)

Reviewer #2: Yes

4. Have the authors made all data underlying the findings in their manuscript fully available?

Reviewer #1: (No Response)

Reviewer #2: Yes

5. Is the manuscript presented in an intelligible fashion and written in standard English?

Reviewer #1: (No Response)

Reviewer #2: Yes

6. Review Comments to the Author

Reviewer #1: (No Response)

Reviewer #2: (No Response)

7. PLOS authors have the option to publish the peer review history of their article (what does this mean?). If published, this will include your full peer review and any attached files.

Reviewer #1: No

Reviewer #2: No

---

## [Editor Report · Acceptance letter]

19 Jun 2023

PONE-D-22-33842R1 

Leukocyte differentiation in bronchoalveolar lavage fluids using higher harmonic generation microscopy and deep learning 

Dear Dr. van Huizen:

I'm pleased to inform you that your manuscript has been deemed suitable for publication in PLOS ONE. Congratulations! Your manuscript is now with our production department. 

Kind regards, 

on behalf of

Dr. Rahul Gomes 

Academic Editor

PLOS ONE